# Buffering Effect of Perseverance and Meditation on Depression among Medical Students Experiencing Negative Family Climate

**DOI:** 10.3390/healthcare10101895

**Published:** 2022-09-28

**Authors:** Nitchamon Pongpitpitak, Nahathai Wongpakaran, Tinakon Wongpakaran, Weerapong Nuansri

**Affiliations:** Department of Psychiatry, Faculty of Medicine, Chiang Mai University, Chiang Mai 50200, Thailand

**Keywords:** family climate, depression, borderline personality symptoms, strengths, meditation, perseverance

## Abstract

Background and objective: Thirty percent of Thai medical students experienced depression. Two of the crucial factors related to depression involved borderline personality disorder symptoms and adverse family experiences, while positive strengths were documented to prevent depression. This study aimed to investigate the role of perseverance and meditation on depression; Methods: Two hundred and forty−three medical student participants completed the following measurements: the core symptom index (CSI−D) to measure depression, the family climate questionnaire to measure family experience, the personality disorder questionnaire to measure borderline personality disorder symptoms, and the inner strength−based inventory to measure perseverance and meditation. In the analysis, depression served as an outcome, adverse family climate as a predictor and borderline personality disorder symptoms as a mediator. In contrast, perseverance and meditation were moderators in the mediation model. Mediation and moderation analysis using PROCESS was applied for testing the direct and indirect effects; Results: Among all, 49.38% were male, and the mean age was 22.76 years. Borderline personality disorder symptoms significantly mediated the relationship between adverse family climate and depression, B = 0.0608 (95%CI, 0.0301 to 0.1052). The variance explained by the mediation model was 43%. Meditation moderated the relationship between adverse family climate and borderline personality symptoms, whereas perseverance and meditation were found to be significant moderators for borderline personality disorder symptoms and depression. With two moderators in the mediation model, the indirect effect index was B = 0.0072 (95%CI, 0.0002 to 0.0160). The 49% of variances of depression were explained by the moderated mediation model; Conclusion: Borderline personality disorder symptoms are a crucial variable involving depression. Meditation practice has been demonstrated to be a buffer between negative family climate to borderline personality disorder symptoms and depressive symptoms, whereas perseverance buffers the effect of borderline personality disorder symptoms of depression. Further research on cultivating meditation and perseverance should be encouraged among those with negative family experiences.

## 1. Introduction

Depression is a common psychiatric disorder leading to a significant adverse event in one’s life, including the possibility of suicidality. Suicide is the second leading cause of death in the teenage group [1]. University students, especially medical students, have strikingly increased in the past 20 years [2,3,4]. A systematic review has reported that the prevalence of depression among university students ranged from 10 to 85% [5]. The prevalence of depression among medical students was found to be higher than that of the general population and age−matched peers [6]. The prevalence of depression among medical students was 28.0% [7]. In Thailand, 30.5% of medical students reported depression, and 12.8% had suicidal ideation during their medical study. Research has shown that depression significantly impacts academic performance [8,9,10].

Depression is related to many factors, including internal factors such as personality traits, low self−esteem and poor problem−solving skills and external factors such as the experience of psychological abuse and negative family intimacy [11]. The experience of growing up in a family was a fundamental and vital step for developing adolescents’ personality traits and subsequent mental health outcomes. Family factors can be described in several ways, e.g., family environment, climate, cohesion and conflict [12]. No consensus has determined the specificity of family influence on individuals. For example, family cohesion, attributable to bonding, dependence and commitment in one individual, is perceived as affecting family climate [13]. However, family environment or climate can be intuitively viewed as positive or negative. Growing up in a positive family climate, where parents exhibit positive emotional interactions, would result in children’s high self−esteem, high stress−coping skills and reduced risks of depression [14]. In contrast, adverse family climates such as poor childcaring, lack of parental support and inability to have positive emotional interactions may contribute to adolescents’ low self−esteem, high levels of stress, poor stress−coping skills and increased risks of depression [15]. Many studies confirmed that a family emotional environment characterized by low maternal warmth and low positive emotion expressiveness significantly predicted depression [16,17,18]. However, the evidence of the impact of family climate on depression among medical students remains lacking.

The most important element related to family climate is, in fact, personality organization. Relevant literature has demonstrated that adverse family climates increased the risk of borderline personality disorder (BPD) symptoms in later years of life [19]. A negative family climate (NFC) was described as involving negative familial relationships [20], negative parenting style, e.g., low warmth, hostility, and punishment [21], less family cohesiveness, more family control and family conflicts [22]. Patients with BPD revealed a high rate of attempting suicide and hospitalization [23]. BPD is associated with other neuroticisms and insecure attachments [24], and BPD was well−documented as a robust predictor of depression and suicidality [25]. Recent research showed a growing number of borderline personality traits among university students, including medical students [26].

While research showed the success of psychotherapy on borderline personality, strength increased from psychotherapy has become the mainstream treatment in helping these people. Meditation was one of the three pillars incorporated in dialectic behavioral therapy and was an example of the importance of cultivating strength for these people. Positive factors can prevent depression when individuals encounter stress [27]. Besides meditation, some clinicians or psychotherapists believe that individuals with BPD might have character strengths such as resilience or hope.

The character strengths commonly studied include cognitive strengths (wisdom), emotional strengths (courage), social and community strengths (humanity and justice), protective strengths (temperance), and spiritual strengths (transcendence) [28]. Another positive characteristic, inner strength, including generosity, perseverance, truthfulness, loving−kindness, wisdom, determination, morality/virtue/precept, patience, equanimity and mindfulness, was also examined [29]. The inner strengths, including meditation, equanimity and precepts, were found to be associated with decreased depression and increased resilience [30,31]. Perseverance and determination were found to be the strongest relationship among all SBI items [32]. Grit is composed of perseverance and passion for long term goals, especially in difficult situations. A related study suggested that high grit can moderate the effect of negative life events on suicidal ideation [33]. Meditation was a positive attribute that purposefully pays attention to the present moment. One investigation on meditation indicated how it affects neuroticism and depression. The study showed that dispositional meditation partially mediated the relationship between neuroticism and depressive symptoms [34,35]. Moreover, it effectively lowers anger and can help moderate affective responses to social rejection among individuals with BPD traits [36]. However, as mentioned, except for meditation, other strengths of BPD are, unfortunately, rarely reported.

Therefore, the study’s objective was to investigate the particular strengths, namely, perseverance and meditation, to see whether they had a moderating effect on the relationship between adverse family climate, borderline personality disorder symptoms and depression among the medical students. The authors hypothesized that at a high level of meditation and perseverance, the effects of an adverse family climate on borderline personality disorder symptoms and depression should be mitigated.

## 2. Materials and Methods

This study employed a cross−sectional design, using paper/pencil and online surveys from 2012 to September 2020 in Thailand. This survey was part of a practicum on a one−month rotation of clinical psychiatry clerkship for fifth−year medical students at the Faculty of Medicine, Chiang Mai University. Convenience and voluntary basis recruitment were applied. No compensation was provided, but a summary was composed of strengths and potential reports of the participants. Estimated sample size calculation was carried out for moderation, and a mediation analysis model using Monte Carlo power analysis for indirect effects. Two parallel mediators with a correlation coefficient of the variables (ranging from 0.156 to 0.320) and the required beta values of 0.8 for indirect effect, yielded a sample size of 83. Ethics review and approval were obtained from the Faculty of Medicine, Chiang Mai University, Thailand.

## 3. Measurements

### 3.1. Core Symptom Index (CSI)

The CSI tool is designed to measure depression, anxiety and somatization symptoms. The CSI instructions direct respondents to answer the items based on how they felt over the past week. The instrument consists of 17 items, all of which are based on a 5−point Likert scale, i.e., values of 0 (never), 1 (rarely), 2 (sometimes), 3 (frequently) and 4 (almost always). Total scores were consistently interpreted; the higher the score, the higher the level of psychopathology [37]. In this study, only the depression subscale was used, and Cronbach’s alpha was 0.856.

### 3.2. Family Climate Questionnaire (FCQ)

The FCQ is an instrument measuring family climate based on 8 dimensions, including the atmospheres of helping, vivacious, fun, enjoyable, sensitive, empathic, overinvolved, controlling, ignoring, distant, chaotic, disorganized, and lonely. It can be grouped into positive and negative family climates. The scale consists of 40 items, 5−Likert scales, ranging from 1 = not at all to 4 = almost always. The FCQ asks respondents to rate their feelings toward their father and mother separately. The total score ranges from the higher score, and a higher level of such quality. The FCQ has 10 items for positive climate with mother (Cronbach = 0.925), 10 items for positive climate with father (Cronbach = 0.938), 10 items for negative climate with mother (Cronbach = 0.917), and 10 items for negative climate with father (Cronbach = 0.915). Therefore, it has 20 items for negative climate for both father and mother. In term of validity, positive family climate was positively related to resilience (r = 0.171, *p* < 0.05), but negatively related to feeling of failure in work (r = −0.143, *p* < 0.05), failure in life (r = −0.145, *p* < 0.05), and failure in relationship (r = −0.153, *p* < 0.05). Conversely, negative family climate was positively related to interpersonal problem (r = 0.137, *p* < 0.05), failure in life (r = 0.143, *p* < 0.05), and failure in relationship (r = 0.132, *p* < 0.05) [38]. An NFC score was used for this analysis.

### 3.3. Personality Assessment: The Diagnostic and Statistical Manual of Mental Disorders (DSM) and the International Classification for Diseases (ICD) Personality Questionnaire (DIP−Q)

DIP-Q assesses personality pathology. The DIP−Q is a self−report questionnaire for screening DSM−IV and ICD−10 Personality disorders, plus schizotypal disorder in ICD−10. This tool consists of 209 true−false statements. It yielded 10 personality traits of DSM, including schizoid, schizotypal, paranoid, narcissistic, antisocial, borderline, histrionic, avoidant, dependent, obsessive−compulsive personality and other personality disorders [39]. This study used only borderline personality disorder symptoms (BPDS). Higher scores indicated more severe symptoms of borderline traits. The Cronbach’s alpha of borderline personality trait was 0.845.

### 3.4. Strength-Based Inventory (SBI)

The ten inner strengths comprised positive psychological characteristics: generosity, morality, mindfulness/meditation, wisdom, perseverance, patience and endurance, truthfulness, determination, loving−kindness and equanimity. The scores of SBI range from 5 to 50. Higher scores reflect those characteristics, and each subscale can be used separately [40]. Meditation and perseverance scores were used in this analysis.

## 4. Data Analysis

Descriptive statistics, e.g., mean, percentage and standard deviation, were used to analyze demographic information such as sex, age, education level, others and scores of CSI, BDS, SBI, FCQ.

To analyze correlations between continuous variables, Pearson’s correlation was used. Spearman’s rank coefficients were used for a category or ordinal variables, and biserial or polyserial were used for nominal and continuous variables.

For multiple regression, the assumption of linearity, homoscedasticity, independence, and normality of residuals was examined using the graphs of standardized residual plots for normal probability and histograms. In mediation analysis in finite samples, however, the total indirect effect is rarely normally distributed. To address this problem, Preacher and Hayes suggested using bootstrapping methods that could be extended to designs involving mediation analysis [41].

To examine how all variables affected the outcome (depression), the mediation model was used in that X represents family climate, Y represents depression, Mediator1 represents borderline symptoms, and Mediator2 represents meditation. To analyze the mediation model, we began by examining the magnitude of the relationships between family climate, depression, meditation and borderline symptoms using zero−order correlations. For mediation analysis, we used the methods discussed by Hayes to examine the relationship between family climate (X) and depression (Y) through borderline symptoms (M). For moderation analysis, the plots were created between NFC (X) and depression (Y), between NFC (X) and BPDS (M), and between BPDS (M) and depression (Y), according to the high and low levels of meditation and perseverance. A significant interaction of each plot was examined by visualizing predicted values of NFC or BPDS with high or low levels of meditation and perseverance [42]. The moderation model demonstrating the existence of moderating effects was included in the full moderated moderation model. According to Hayes [42], when the moderation effect existed at a, b, and c, then seven moderated mediation models were possibly created; therefore, each model would be tested. To produce more accurate results of mediation and moderation analysis, 5000 resampling or bootstrapping methods were applied [41,42]. The results were reported by unstandardized estimates, bootstrapped standard errors, and bootstrapped confidence intervals for conditional direct ad indirect effects. Confidence intervals that do not include zero are indicative of statistical significance. For all the analyses, the level of significance was set at *p* < 0.05. All statistical analyses were performed using the IBM SPSS Program, 22.0. and PROCESS macro, Version 4.1 annexed to IBM SPSS was used for all mediation and moderation analyses. MedCalc, Version 19.7 was used to create scatter plots and regression lines.

## 5. Results

Among all participants, one half were female. The mean and standard deviation of the measurement scores is shown in Table 1, and Table 2 shows the correlation coefficients between each pair of variables. As expected, NFC was positively related to BPDS and depression, while perseverance and meditation were negatively related to BPDS and depression (*p* < 0.01). Surprisingly, NFC was positively associated with meditation.

In exploring the moderation effect of perseverance, significant interaction term between NFC and perseverance on depression was observed (*β* = −0.064, *p* <0.01) (Figure 1), but not on BPDS (*β* = 0.019, *p* = 0.301). However, the interaction term between BPDS and perseverance on depression was significant (*β* = −0.35287512, *p* < 0.001) (Figure 2).

In exploring the moderation effect of meditation, significant interaction term between NFC and meditation on BPDS was observed (*β* = −0.035, *p* < 0.001) (Figure 3), likewise on depression (*β* = 0.073, *p* < 0.001) (Figure 4). However, the interaction term between BPDS and meditation on depression was significant (*β* = −0.012, *p* = 0.881).

Based on the correlation among variables and the moderating effects depicted by graphical illustrations, we hypothesized the moderation and mediation model 22 according to Hayes [42] (Figure 5). Meditation practice was entered as a moderator of the relationship between NFC and borderline personality symptoms, perseverance was entered as a moderator of the relationship between borderline personality disorder symptoms and depression, and borderline personality disorder symptoms were entered as a mediator of the relationship between NFC and depression. Depression was the dependent variable (Figure 5). The analysis assessed (1) the effects of NFC tendency on depression (both directly and indirectly, through borderline personality symptoms), (2) the effect of NFC on borderline personality disorder symptoms (as moderated by meditation), (3) the effect of NFC on depression (as moderated by meditation) and (4) the effect of borderline personality disorder symptoms on depression (as moderated by perseverance).

Figure 5 shows the direct effect of NFC (B = 0.0798, 95% CI = 0.0377 to 0.123, *p* < 0.001), BPDS (B = 0.7688, 95% CI = 0.577 to 0.9727, *p* < 0.001), Meditation (B = −0.38, 95% CI = −0.752 to −0.0081, *p* < 0.05), Interaction effect of NFC and Meditation (B = −0.0371, 95% CI = −0.0581 to −0.0148, *p* < 0.05), Perseverance (B = 0.0798, 95% CI = −0.5928 to 0.2761, *p* > 0.05), Interaction effect of BPDS and perseverance (B = −0.2129, 95% CI = −0.4075 to −0.0154, *p* < 0.05),and Indirect effect index (B = 0.0072, 95% CI = 0.0002 to 0.0160, *p* < 0.05) on depression.

Table 3 shows the direct and indirect effects of the NFC, borderline personality symptoms, meditation and perseverance on depression. Model A: NFC predicted depression at approximately 8% and increased to 43% when borderline personality disorder symptoms were added as a mediator. The direct effect of NFC reduced from 0.094 to 0.0364. The indirect effect of NFC on depression through borderline personality disorder symptoms was 0.0608 (95%CI, 0.0301 to 0.1052) (Model B). Model C1 and C2 illustrated the moderated mediation models. Model C1 displayed a significant indirect effect index, B = −0.0290 (95%CI, −0.0451 to −0.0112), when meditation served as a moderator, whereas model C2, a significant indirect effect index, B = −0.0160 (95%CI, −0.0378 to −0.0032), when perseverance served as a moderator. Model D showed the results of the moderated mediated model when the mediator was moderated by two moderators− meditation and perseverance. The indirect effect index was B = 0.0072 (95%CI, 0.0002 to 0.0160). The variance of depression was explained by 49% by the predictors in the final model.

Table 4 presents the results showing the index of moderated mediation was significant [95 % Cl: = 0.00 (0.00)], indicating that the indirect effect of NFC on depression among medical students through meditation was moderated by meditation and perseverance. Table 4 also presents the conditional indirect effect on values of the moderators calculated based on one standard deviation above (+SD) and one standard deviation below (−SD). The results revealed significant indirect effects of levels of meditation; low meditation (β = 0.1106, *p* < 0.001), 95% Cl: = 0.0708 to 0.1536), moderate meditation (β = 0.0831, *p* < 0.001), 95% Cl: = 0.0552 to 0.1154) and high meditation (β = 0.0556, *p* < 0.05), 95 % Cl: = 0.036 to 0.0866). The result also showed significant indirect effects of levels of perseverance; low perseverance (β = 0.0691, *p* < 0.001), 95% Cl: = 0.0338 to 0.1201), moderate perseverance (β = 0.0543, *p* < 0.001), 95% Cl: = 0.0268 to 0.0934) and high perseverance (β = 0.0394, *p* < 0.001), 95% Cl: = 0.0164 to 0.0736). Thus, the indirect effects of NFC on depression among individuals with diabetes through borderline personality disorder symptoms and the direct effect moderation can be achieved through low, moderate and high levels of meditation and perseverance among medical students.

Table 5 shows the combination between two moderators, meditation and perseverance, in predicting depression. All combinations yielded significant indirect effects. For example, a low level of meditation and low level of perseverance yielded 0.1305 (95%CI 0.0836 to 0.1829), whereas a high level of meditation and low level of perseverance yielded 0.0634 (95%CI 0.0408 to 0.0975).

## 6. Discussion

The present study aimed at testing the buffering effect of meditation and perseverance on depression. Consistent with related studies, NFC increased the risks of depression [15] and meditation practice buffered depression [43,44]. In contrast to other related studies that investigated the meditating effect of meditation practice [31,34,35], the present study demonstrated the significant buffering effect of meditation on NFC on depression, indicating that a higher level of meditation practice indicated a lower level of depression would be observed. The same is true for borderline personality symptoms. A higher level of meditation practice indicated a lower level of borderline symptoms. This finding was endorsed by a related study in that the effortful control of mindfulness meditation was lower among individuals with borderline personality disorder symptoms [45]. The fact that meditation has two buffering effects on borderline personality disorder symptoms and depression may be related to emotional control and regulatory practices. Meditation may also help individuals growing up in such negative family environments to deal with negative thoughts and beliefs that are prone to subsequent depression [46,47,48]. Likewise, buffering effect of meditation on NFC concerning borderline personality disorder symptoms endorses the related study, showing that meditation moderates affective responses to social rejection among individuals with borderline personality traits [36].

On the other hand, perseverance was found to be a buffer for depression among those with borderline personality symptoms. Interestingly, these two moderators serve different functions in the model mediation model in that meditation practice suggested having a lower likelihood of borderline personality symptoms, and perseverance practice suggested having a lower likelihood of depression. In line with other related studies, albeit with different study outcomes, one study demonstrated that perseverance could moderate the effect of negative life events on suicidal ideation [33]. The moderating effect of perseverance and borderline personality disorder symptoms of depression suggests that in certain situations, individuals are able to cultivate perseverance attributes.

Based on this proposed moderated mediation model, the authors believe this is probably the first study on the outcomes of depressive symptoms among medical students presenting borderline personality disorder symptoms and experiencing NFC in life.

As mentioned earlier, inner or character strengths can be cultivated in everyone regardless of what mental health problem he/she experiences. Positive emotions or psychological aspect is not the reduced negative mental health experience but a different entity that should be integrated [49]. Moreover, any strength does not stand alone; rather, it affects or relies on other strengths to present itself, as evidenced by meditation and perseverance are significantly related [50].

The strengths that would prevent depression can be derived from various methods. Commonly, patients may learn them in psychotherapeutic situations. However, individuals may gain such strengths from many resources in daily life. For example, junior medical students at the Faculty of Medicine, Chiang Mai University, are promoted and provided to have an opportunity to meditation practice as part of extra−curricular activity [51,52].

In Thailand, where Theravada Buddhism has long been securely based, meditation practice is omnipresent and considered a practical method to enhance mental well-being among Thais and foreigners. Access to meditation practice is easy and usually free of charge. Formal meditation courses are accessible, and nonformal meditation is promoted all over the country and nearly every month through Buddhist or national holidays, e.g., Vesak, Magha Puja, Asaha Puja, Buddhist Lent and the King and Queen’s birthdays. That is why medical students have the opportunity to receive meditation and other strength practices.

However, the present findings can be applied to any culture as meditation is deemed universal. Adolescents and young adults, especially individuals who grew up experiencing NFC, should have an opportunity to be involved with positive emotions like meditation and other strengths. Notably, medical students who will be future doctors and determined to serve people facing challenges should have an opportunity to develop a robust psychological immunity from such inner strengths.

### Strengths and Limitation

This study may be one of the early studies to demonstrate the effect of meditation and perseverance on depressive symptoms. However, some limitations should be mentioned here. First, this study’s results could only assess the symptoms and depression through self−report questionnaires, so the findings might not be entirely representative of those with borderline personality or depressive disorders. In addition, social desirability bias from self−report may be inevitable. Therefore, the interpretation of the results should be cautious. Second, our data cannot exclude participants presenting major depressive disorder or currently receiving psychological treatment, which may have influenced decreased depressive symptoms and increased strength due to therapy. Finally, this cross−sectional design cannot confirm causal inference; a longitudinal design should be encouraged.

## 7. Conclusions

Meditation and perseverance are crucial positive strengths to minimize the risk of depression symptoms. Meditation has a significant buffering effect on depression among those who grew up experiencing adverse family climates. Perseverance also buffers depression among those with borderline personality symptoms.

## Figures and Tables

**Figure 1 healthcare-10-01895-f001:**
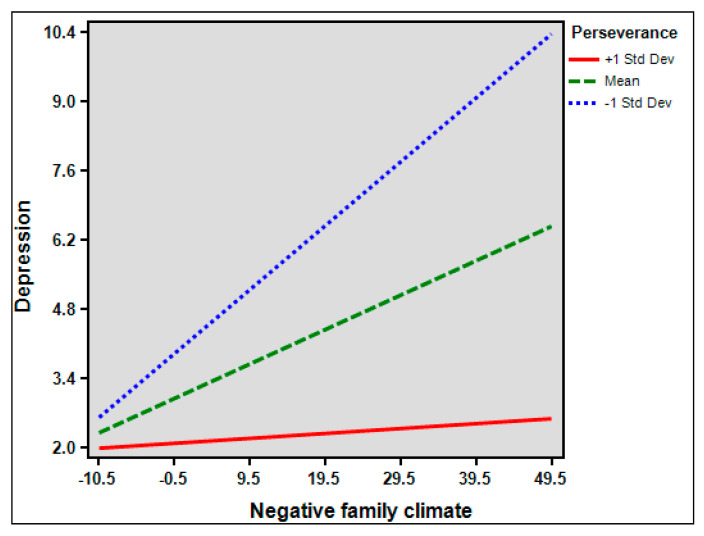
Depression scores as a function of the interaction between Negative Family Climate and perseverance.

**Figure 2 healthcare-10-01895-f002:**
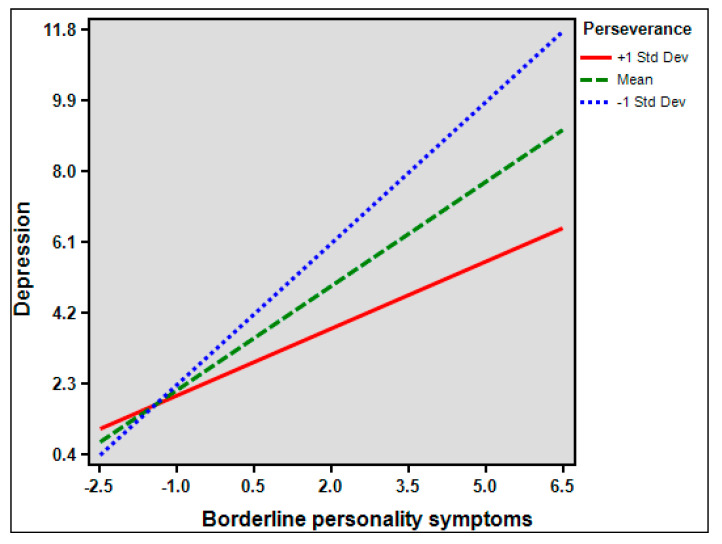
Depression scores as a function of the interaction between borderline personality disorder symptoms and perseverance.

**Figure 3 healthcare-10-01895-f003:**
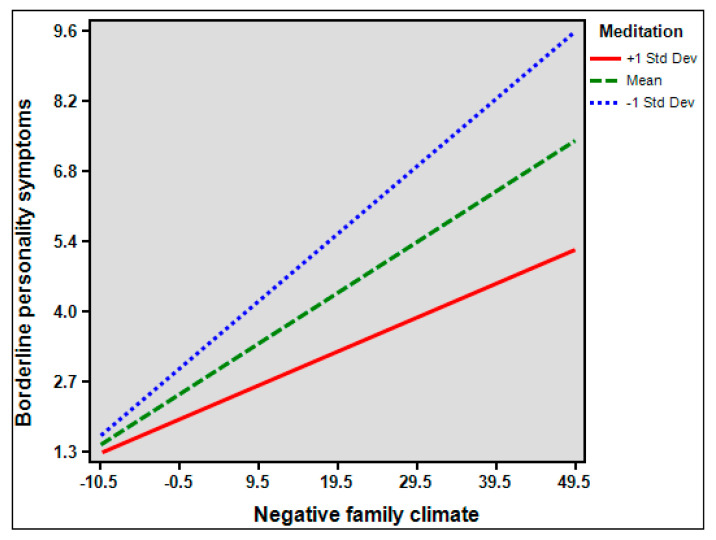
Borderline personality disorder symptoms as a function of the interaction between Negative Family Climate and meditation.

**Figure 4 healthcare-10-01895-f004:**
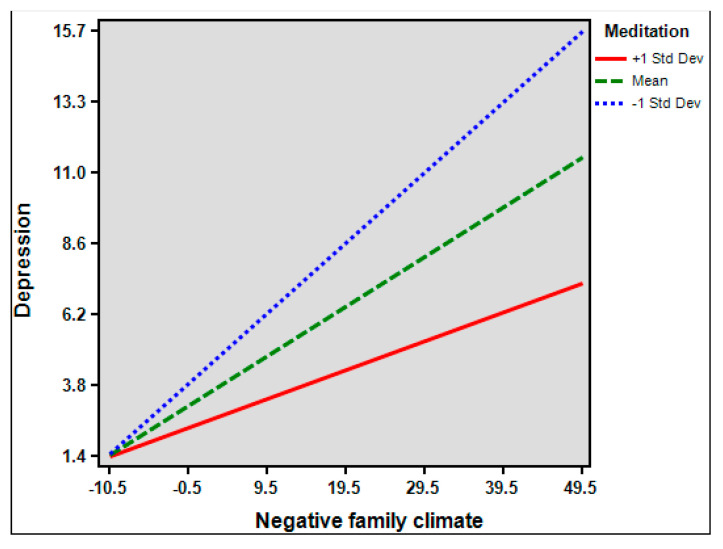
Depression scores as a function of the interaction between Negative Family Climate and meditation.

**Figure 5 healthcare-10-01895-f005:**
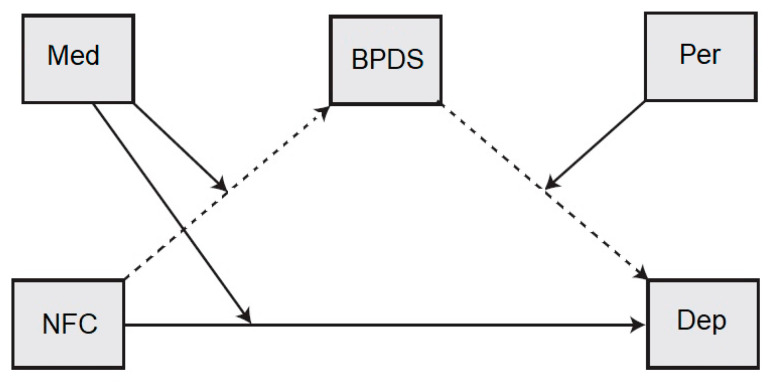
The multiple moderated mediation analysis (Model 22). BPDS = borderline personality disorder symptoms, NFC = negative family climate, Dep = Depression, Med = meditation, Per = Perseverance, Single-headed arrow from the variable in rectangular boxes represents path coefficients or direct effect. Solid arrow indicates moderation; Dashed arrow indicates mediation, and indirect pathway. Solid lines are direct effect; Dot lines are indirect effect.

**Table 1 healthcare-10-01895-t001:** Sociodemographic and clinical data.

Variables	n (%) or Mean ± SD
Age (year)	22.76 ± 0.91
Sex, female	123 (50.62)
Perseverance	2.93 ± 0.95
Meditation	2.09 ± 1.04
Depression	3.32 ± 3.95
Borderline Personality Symptoms	2.51 ± 2.37
Negative Family Climate	31.44 ± 12.67
Perseverance level	
High	151 (62.13)
Low	92 (37.86)
Meditation level	
High	166 (68.31)
Low	77 (31.69)

**Table 2 healthcare-10-01895-t002:** Zero order correlation between variables.

	Sex	Perseverance	Meditation	Depression	BPDS	NFC
Age	−0.044	−0.026	0.100	−0.126	−0.095	0.173 **
Sex	−	0.161 *	−0.012	−0.070	−0.118	−0.138 *
Perseverance		−	0.267 **	−0.320 **	−0.326 **	−0.227 **
Meditation			−	−0.230 **	−0.182 **	0.302 **
depression				−	0.632 **	0.150 *
BPDS					−	0.188 **

BPDS = borderline personality disorder symptoms, NFC = negative family climate. * *p* < 0.05, ** *p* < 0.01.

**Table 3 healthcare-10-01895-t003:** Summary of moderated mediation analysis of NFC, meditation and perseverance predicting depression.

		Y (Depression)				
Model	Predictor (X)	Unstandardized Coefficient (B)	SE	BootLLCI	BootULCI	R²
A.Regression	Constant	0.297	1.044	−2.208	1.895	0.083
	NFC	0.094 ***	0.037	0.037	0.183	
B. Model 4	Constant	−0.2621	0.6015	−1.6102	0.7570	
	NFC	0.0364	0.0226	0.0011	0.0886	0.4332
	BPSD	0.9487 ***	0.0913	0.7744	1.1344	
	Indirect effect	0.0608 *	0.0193	0.0295	0.1061	
C1. Model8	Constant	1.1781 **	0.2057	0.7872	1.5852	
	NFC	0.084 ***	0.0217	0.0411	0.1275	0.4770
	BPSD	0.8315 ***	0.0952	0.6496	1.0207	
	Meditation	−0.3722 *	0.1776	−0.7248	−0.015	
	NFC×Meditation	−0.0442 ***	0.0107	−0.0649	−0.0238	
	Indirect effect index	−0.0290 *	0.0090	−0.0451	−0.0112	
C2. Model 14	Constant	1.8518	0.6673	0.3514	2.9822	
	NFC	0.0366 *	0.0221	0.0016	0.0873	0.4580
	BPSD	0.8477 ***	0.0963	0.6594	1.0327	
	Perseverance	−0.3044	0.2266	−0.7468	0.1485	
	BPD×erseverance	−0.2533 **	0.0966	−0.4445	−0.0638	
	Indirect effect index	−0.0160 *	0.0088	−0.0378	−0.0032	
D. Model 22	Constant	3.0437 **	0.1871	2.6862	3.4223	
	NFC	0.0798 ***	0.022	0.0377	0.123	0.4913
	BPSD	0.7688 ***	0.1008	0.577	0.9727	
	Meditation	−0.38 *	0.1868	−0.752	−0.0081	
	NFC×Meditation	−0.0371**	0.0117	−0.0581	−0.0148	
	Perseverance	−0.164	0.2217	−0.5928	0.2761	
	BPSD×Perseverance	−0.2129 *	0.0991	−0.4075	−0.0154	
	Indirect effect index	0.0072 *	0.0042	0.0002	0.0160	

BPDS = borderline personality disorder symptoms, NFC = negative family climate, Dep = Depression, SE = standard error, BootLLCI = bootstrapped lower level confidence interval, BootULCI = bootstrapped upper level confidence interval, R² = r-square, * *p* < 0.05, ** *p* < 0.01 *** *p* < 0.001.

**Table 4 healthcare-10-01895-t004:** Summary of analysis of conditional indirect effect of NFC on depression.

	Meditation				Perseverance			
	B	BootSE	BootLLCI	BootULCI	B	BootSE	BootLLCI	BootULCI
Low (−1SD)	0.1106	0.0213	0.0708	0.1536	0.0691	0.0223	0.0338	0.1201
Moderate	0.0831	0.0154	0.0552	0.1154	0.0543	0.017	0.0268	0.0934
High (+1SD)	0.0556	0.013	0.036	0.0866	0.0394	0.0148	0.0164	0.0736

SD = standard deviation, NFC = negative family climate, B = unstandardized coefficient, BootLLCI = bootstrapped lower level confidence interval, BootULCI = bootstrapped upper level confidence interval, R² = r square.

**Table 5 healthcare-10-01895-t005:** Conditional indirect effect of family climate on depression through borderline personality disorder symptoms was moderated by meditation and perseverance.

Meditation	Perseverance	Effect	BootSE	BootLLCI	BootULCI
Low	Low	0.1277	0.025	0.0804	0.1793
Low	Moderate	0.1022	0.0204	0.0632	0.1442
Low	High	0.0767	0.0235	0.0322	0.1253
Moderate	Low	0.096	0.0181	0.0629	0.1348
Moderate	Moderate	0.0768	0.0147	0.0499	0.1081
Moderate	High	0.0576	0.0174	0.025	0.0935
High	Low	0.0642	0.0152	0.0415	0.0986
High	Moderate	0.0514	0.0121	0.033	0.0788
High	High	0.0386	0.0128	0.0168	0.0671

BootLLCI = bootstrapped lower level confidence interval, BootULCI = bootstrapped upper level confidence interval.

## Data Availability

The datasets used and/or analyzed during the current study are available from the corresponding author upon reasonable request.

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
