# Peer review of "Buffering Effect of Perseverance and Meditation on Depression among Medical Students Experiencing Negative Family Climate"

_healthcare, 2022, doi:10.3390/healthcare10101895_

Round 1
Reviewer 1 Report
This article by Nitchamon Pongpitpitak et al aims to investigate the buffering effects of perseverance and meditation on depression in medical students who experience a negative family atmosphere. Overall, the paper is well written, but still needs some minor revisions.
The manuscript is relatively well-written and well-structured. I enjoy reading it; however, some parts of the manuscript are not described in full detail, and readers would benefit from a more comprehensive description. Please find my comments, I would be grateful if the author addresses them.
1. Line 17. Recheck the grammar of this sentence
2. Line 27. “variances … was”?
3. Line 32. “experience”?
4. Please check not to omit any abbreviations in the text, including the abstract.
5. Line 37. “significantly” or “significantly”?
6. Although the text is easy to understand, the English version needs to be improved.
7. P-value or P
8. Fig5. Can you adjust the font size?
9. Table4. Whether the layout can be adjusted
10. The author mentioned in the Discussion that there are subjective factors in the form of the questionnaire
Author Response
Dear Editor and reviewers,
We thank for your valuable comments to improve our manuscript. Please see below our responses to your comments.
Reviewer 1
This article by Nitchamon Pongpitpitak et al aims to investigate the buffering effects of perseverance and meditation on depression in medical students who experience a negative family atmosphere. Overall, the paper is well written, but still needs some minor revisions.
The manuscript is relatively well-written and well-structured. I enjoy reading it; however, some parts of the manuscript are not described in full detail, and readers would benefit from a more comprehensive description. Please find my comments, I would be grateful if the author addresses them.
Response. Thank you very much for your positive feedback.
- Line 17. Recheck the grammar of this sentence
Response: We have already checked and corrected it. It now reads “In the analysis, depression served as an outcome, adverse family climate as a predictor and borderline personality symptoms as a mediator. In contrast, perseverance and meditation were moderators in the mediation model.”
- Line 27. “variances … was”?
Response: Thank you. We have corrected it. It now reads
“The 49% of variances of depression were explained by the moderated mediation model”
- Line 32. “experience”?
Response: Thank you. We have corrected it. It now reads “Further research on cultivating meditation and perseverance should be encouraged among those with negative family experiences”
- Please check not to omit any abbreviations in the text, including the abstract.
Response: We have already checked and corrected it throughout the manuscript and abstract
- Line 37. “significantly” or “significantly”?
Response: It should be “significant”. Thank you we have revised it accordingly.
- Although the text is easy to understand, the English version needs to be improved.
Response: Thank you. We have checked the whole our manuscript again.
- P-value or P
Response: We have already checked and corrected it to P
- Can you adjust the font size?
Response: We have already corrected it as suggested.
- Table 4. Whether the layout can be adjusted
Response: We have already checked and corrected.
- The author mentioned in the Discussion that there are subjective factors in the form of the questionnaire
Response: Yes, we see that as limitation when using self-report questionnaires. However, as mental health outcomes (e.g., depression) rely heavily on subjective report rather than objective or hard evidence (e.g., blood test, brain scan), we believe that using self-report questionnaire still be useful and appropriate.
We hope we could make our revised manuscript more clarified. We are looking forward to hearing from you.
Thank you very much.
Best regards,
TW

Reviewer 2 Report
This is a interesting article in the context of Mental Health.
However, the following recommendations should be taken into account by
the authors and assess their relevance:
1. The aim of the study is not clearly established.
2. In material and method, the authors indicate that the surveys were carried out from 2012 to 2020. However, the ethics committee report is from 2017. Could you clarify why the ethics report is after data collection?
3. It would be convenient to explain more precisely Figure 5 in the results
4. In the last paragraph of the discussion, the authors seem to indicate statements corresponding to the conclusion. Please review or clarify
Author Response
Dear Editor and reviewers,
We thank for your valuable comments to improve our manuscript. Please see below our responses to your comments.
Reviewer 2
This is a interesting article in the context of Mental Health.
However, the following recommendations should be taken into account by
the authors and assess their relevance:
- The aim of the study is not clearly established.
Response: We have already revised as suggested
Therefore, the study's objective was to investigate the particular strengths, namely, perseverance and meditation, to see whether they had a moderating effect on the relationship between adverse family climate, borderline personality symptoms and depression among the medical students. The authors hypothesized that at a high level of meditation and perseverance, the effects of an adverse family climate on borderline personality symptoms and depression should be mitigated.
- In material and method, the authors indicate that the surveys were carried out from 2012 to 2020. However, the ethics committee report is from 2017. Could you clarify why the ethics report is after data collection?
Response: We apologize for the mistake. We already correct the data.
- It would be convenient to explain more precisely Figure 5 in the results
Response: We have added the result to explain figure 5 as followed
- Figure 5 representing Model 22
- BPDS = borderline personality disorder symptoms, NFC = negative family climate, Dep = Depression, Med = meditation, Per = Perseverance, Single-headed arrow from the variable in rectangular boxes represents path coefficients or direct effect. Solid arrow indicates moderation; Dashed arrow indicates mediation, and indirect pathway.
- The figure shows the direct effect of NFC (B = 0798)(95% CI = 0.0377), BPDS (B = 0.7688)(95% CI = 0.577), Meditation(B = −0.38)(95% CI = −0.752), Interaction effect of NFC and Meditation (B = −0.0371)(95% CI = −0.0581), Perseverance(B = 0.0798)(95% CI = −0.5928), Interaction effect of BPDS and perseverance (B = −0.2129)(95% CI = −0.4075),and Indirect effect index (B = 0.0072)(95% CI = 0.0002) on depression
- In the last paragraph of the discussion, the authors seem to indicate statements corresponding to the conclusion. Please review or clarify
Response: We have revised this part by combining this paragraph with conclusion
We hope we could make our revised manuscript more clarified. We are looking forward to hearing from you.
Thank you very much.
Best regards,
TW

Reviewer 3 Report
See uploaded document

Author Response
Dear Editor and reviewers,
We thank for your valuable comments to improve our manuscript. Please see below our responses to your comments.
Reviewer 3
Response: Thank you for these comments. We agree with that, therefore, we have revised this part as follows.
For multiple regression, the assumption of linearity, homoscedasticity, independence, and normality of residuals was examined using the graphs of standardized residual plots for normal probability and histograms. In mediation analysis in finite samples, however, the total indirect effect is rarely normally distributed. To address this problem, Preacher and Hayes suggested using bootstrapping methods could be extended to designs involving mediation analysis[1].
Response: Thank you for your recommendation. In fact, our research follows the research onion in that
First layer: Interpretivism emphasizes the influence that social and cultural factors can have on an individual: Our study hypothesized that family climate is important for individual’s growing up. Human strength is another philosophy.
Second layer: We approach with deduction by building the model that negative family climate predicts borderline personality disorder symptoms and depression, while positive strengths buffer borderline personality disorder symptoms and depression.
Third layer: To test our hypothesis, we used monomethod quantitative research
Fourth layer: survey was applied in this study.
Fifth layer: cross-sectional
Sixed layer: Data collection and Data analysis
In all, we believe that our manuscript has addressed all the six layers in the regular structured format.
Response: We have revised this scale to be more clarified.
The FCQ has 10 items for positive climate with mother (Cronbach = 0.925), 10 items for positive climate with father (Cronbach = 0.938), 10 items for negative climate with mother (Cronbach = 0.917), and 10 items for negative climate with father (Cronbach = 0.915). Therefore, it has 20 items for negative climate for both father and mother. In term of validity, positive family climate was positively related to resilience (r = 0.171, p<.05), but negatively related to feeling of failure in work (r = -0.143, p<.05), failure in life (r = -0.145, p<.05), and failure in relationship (r = -0.153, p<.05). Conversely, negative family climate was positively related to interpersonal problem (r = 0.137, p<.05), failure in life (r = 0.143, p<.05), and failure in relationship (r = 0.132, p<.05)
We hope we could make our revised manuscript more clarified. We are looking forward to hearing from you.
Thank you very much.
Best regards,
TW

Round 2
Reviewer 2 Report
Dear authors
Thanks for the clarifications provided.
In the second question about the ethics committee, you confirm that you have changed the data, however, I am not able to see said change. Could you explain to me why they started collecting data in 2012 and the ethics committee approved the work in 2017?
Regards,
Author Response
23 September 2022
Dear Editor and reviewers,
We thank for your further thoughtful comments to improve our manuscript. Please see below our responses to your comments.
Reviewer 1
Query
In the second question about the ethics committee, you confirm that you have changed the data, however, I am not able to see said change. Could you explain to me why they started collecting data in 2012 and the ethics committee approved the work in 2017?
Response:
We apologize for the unclear answer last time. The survey was part of a practicum on a one−month rotation of clinical psychiatry clerkship of fifth−year medical students at the Faculty of Medicine, Chiang Mai University. Data had been collected since 2012. However, when the authors wanted to use this database for this research, we applied for Ethical approval. The date submitted for this research and received the approval was 30 August 2021. We have corrected the date on the manuscript in pink color.
Reviewer 2
Query
In lines 260-265, the authors give many beta coefficients and confidence intervals, but without the p-values reported next to the beta coefficients, there are almost useless. A coefficient reported with a p-value to show whether it's statistically significant or not is much more valuable than a coefficient just reported with confidence intervals (that may even be statistically insignificant!).
Response :
Thank you for your suggestion. We have added the p-values for each coefficient. Also, we have revised Table 3, adding the asterisk (*) to denote the level of significance, i.e., * p <0.05, ** p<0.01, *** p < 0.001. The revised texts are in pink colour.
We hope we can make our revised manuscript more clarified. Nevertheless, again, we are looking forward to hearing from you.
Thank you very much.
Best regards,
TW

Reviewer 3 Report
All my comments in the previous round were adequately addressed, however, the authors should look into this comment:
In lines 260-265, the authors give many beta coefficients and confidence intervals, but without the p-values reported next to the beta coefficients, there are almost useless. A coefficient reported with a p-value to show whether it's statistically significant or not is much more valuable than a coefficient just reported with confidence intervals (that may even be statistically insignificant!).
Author Response

(The authors gave the same response as above.)
